# Automatically Composing Representation Transformations as a Means for Generalization

**Michael B. Chang**
Electrical Engineering and Computer Science
University of California, Berkeley, USA
mbchang@berkeley.edu

**Abhishek Gupta**
Electrical Engineering and Computer Science
University of California, Berkeley, USA
abhigupta@berkeley.edu

**Sergey Levine**
Electrical Engineering and Computer Science
University of California, Berkeley
svlevine@eecs.berkeley.edu

**Thomas L. Griffiths**
Psychology and Cognitive Science
Princeton University, USA
tomg@princeton.edu

## ABSTRACT

A generally intelligent learner should generalize to more complex tasks than it has previously encountered, but the two common paradigms in machine learning – either training a separate learner per task or training a single learner for all tasks – both have difficulty with such generalization because they do not leverage the compositional structure of the task distribution. This paper introduces the *compositional problem graph* as a broadly applicable formalism to relate tasks of different complexity in terms of problems with shared subproblems. We propose the *compositional generalization problem* for measuring how readily old knowledge can be reused and hence built upon. As a first step for tackling compositional generalization, we introduce the *compositional recursive learner*, a domain-general framework for learning algorithmic procedures for composing representation transformations, producing a learner that reasons about what computation to execute by making analogies to previously seen problems. We show on a symbolic and a high-dimensional domain that our compositional approach can generalize to more complex problems than the learner has previously encountered, whereas baselines that are not explicitly compositional do not.

## 1 INTRODUCTION

This paper seeks to tackle the question of how to build machines that leverage prior experience to solve more complex problems than they have previously encountered. How does a learner represent prior experience? How does a learner apply what it has learned to solve new problems? Motivated by these questions, this paper aims to formalize the idea of, as well as to develop an understanding of the machinery for, *compositional generalization* in problems that exhibit compositional structure. The solutions for such problems can be found by composing in sequence a small set of reusable partial solutions, each of which tackles a subproblem of a larger problem. The central contributions of this paper are to frame the shared structure across multiple tasks in terms of a *compositional problem graph*, propose *compositional generalization* as an evaluation scheme to test the degree a learner can apply previously learned knowledge to solve new problems, and introduce the *compositional recursive learner*, a domain-general framework[1] for sequentially composing representation transformations that each solve a subproblem of a larger problem.

The key to our approach is recasting the problem of generalization as a problem of learning algorithmic procedures over representation transformations. A solution to a (sub)problem is a transformation between its input and output representations, and a solution to a larger problem composes

---

[1]https://github.com/mbchang/crl

these subsolutions together. Therefore, representing and leveraging prior problem-solving experience amounts to learning a set of reusable primitive transformations and their means of composition that reflect the structural properties of the problem distribution.

This paper introduces the compositional recursive learner (CRL), a framework for learning both these transformations and their composition together with sparse supervision, taking a step beyond other approaches that have assumed either pre-specified transformation or composition rules (Sec. 5). CRL learns a modular recursive program that iteratively re-represents the input representation into more familiar representations it knows how to compute with. In this framework, a transformation between representations is encapsulated into a *computational module*, and the overall program is the sequential combination of the inputs and outputs of these modules, whose application are decided by a *controller*.

What sort of training scheme would encourage the spontaneous specialization of the modules around the compositional structure of the problem distribution? First, exposing the learner to a diverse distribution of compositional problems helps it pattern-match across problems to distill out common functionality that it can capture in its modules for future use. Second, enforcing that each module have only a local view of the global problem encourages task-agnostic functionality that prevents the learner from overfitting to the empirical training distribution; two ways to do this are to constrain the model class of the modules and to hide the task specification from the modules. Third, training the learner with a curriculum encourages the learner to build off old solutions to solve new problems by re-representing the new problem into one it knows how to solve, rather than learning from scratch.

How should the learner learn to use these modules to exploit the compositional structure of the problem distribution? We can frame the decision of which computation to execute as a reinforcement learning problem in the following manner. The application of a sequence of modules can be likened to the execution trace of the program that CRL automatically constructs, where a computation is the application of a module to the output of a previous computation. The automatic construction of the program can be formulated as the solution to a sequential decision-making problem in a meta-level Markov decision process (MDP) (Hay et al., 2014), where the state space is the learner's internal states of computation and the action space is the set of modules. Framing the construction of a program as a reinforcement learning problem allows us to use techniques in deep reinforcement learning to implement loops and recursion, as well as decide on which part of the current state of computation to apply a module, to re-use sub-solutions to solve a larger problem.

Our experiments on solving multilingual arithmetic problems and recognizing spatially transformed MNIST digits (LeCun et al., 1998) show that the above proposed training scheme prescribes a type of *reformulation*: re-representing a new problem in terms of other problems by implicitly making an *analogy* between their solutions. We also show that our *meta-reasoning* approach for deciding what modules to execute achieves better generalization to more complex problems than monolithic learners that are not explicitly compositional.

## 2   COMPOSITIONAL GENERALIZATION

> *Solving a problem simply means representing it so as to make the solution transparent.*
>
> (SIMON, 1988)

Humans navigate foreign cities and understand novel conversations despite only observing a tiny fraction of the true distribution of the world. Perhaps they can extrapolate in this way because the world contains compositional structure, such that solving a novel problem is possible by composing previously learned partial solutions in a novel way to fit the context.

With this perspective, we propose the concept of *compositional generalization*. The key assumption of compositional generalization is that harder problems are composed of easier problems. The problems from the training and test sets share the same primitive subproblems, but differ in the manner and complexity with which these subproblems are combined. Therefore, problems in the test set can be solved by combining solutions learned from the training set in novel ways.

**Definition.** Let a *problem* $P$ be a pair $(X_{in}, X_{out})$, where $X_{in}$ and $X_{out}$ are random variables that respectively correspond to the input and output representations of the problem. Let the distribution

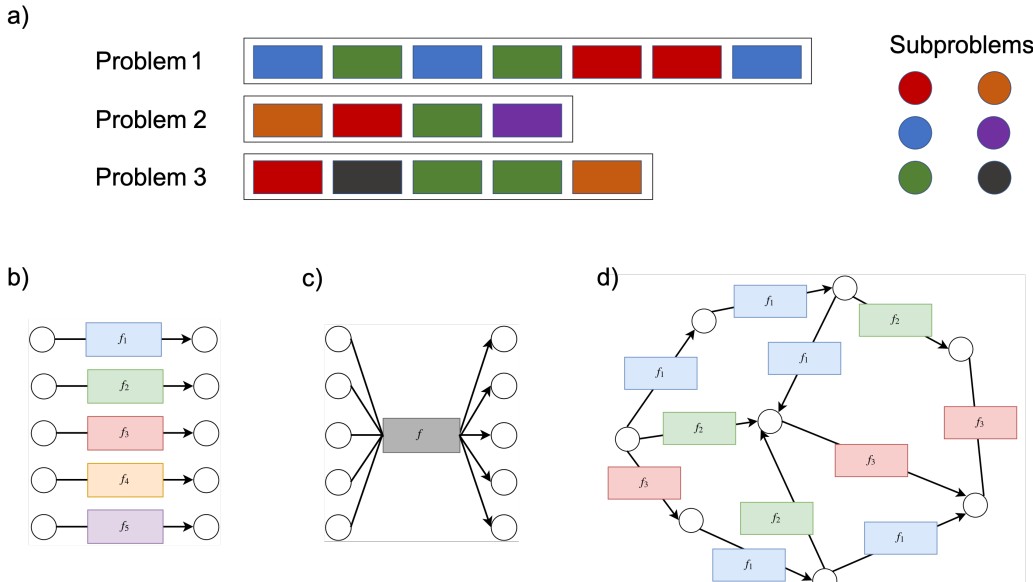

Figure 1: **(a)** Consider a multitask family of problems, whose subproblems are shared within and across problems. Standard approaches either **(b)** train a separate learner per task or **(c)** train a single learner for all tasks. Both have difficulty generalizing to longer compositional problems. **(d)** Our goal is to re-use previously learned sub-solutions to solve new problems by composing computational modules in new ways.

of $X_{in}$ be $r_{in}$ and the distribution of $X_{out}$ be $r_{out}$. To solve a particular problem $P = p$ is to transform $X_{in} = x_{in}$ into $X_{out} = x_{out}$. A composite problem $p_a = p_b \circ p_c$ is that for which it is possible to solve by first solving $p_c$ and then solving $p_b$ with the output of $p_c$ as input. $p_b$ and $p_c$ are subproblems with respect to $p_a$. The space of compositional problems form a *compositional problem graph*, whose nodes are the representation distributions $r$. A problem is described as pair of nodes between which the learner must learn to construct an edge or a path to transform between the two representations.

**Characteristics.** First, there are many ways in which a problem can be solved. For example, translating an English expression to a Spanish one can be solved directly by learning such a transformation, or a learner could make an *analogy* with other problems by first translating English to French, and then French to Spanish as intermediate subproblems. Second, sometimes a useful (although not only) way to solve a problem is indicated by the *recursive* structure of the problem itself: solving the arithmetic expression $3 + 4 \times 7$ modulo 10 can be decomposed by first solving the subproblem $4 \times 7 = 8$ and then $3 + 8 = 1$. Third, because a problem is just an (input, output) pair, standard problems in machine learning fit into this broadly applicable framework. For example, for a supervised classification problem, the input representation can be an image and the output representation a label, and intermediate subproblems can be transforming some intermediate representations to other intermediate representations. Sec. 4 demonstrates CRL on all three of the above examples.

**Broad Applicability.** Problems in supervised, unsupervised, and reinforcement learning can all be viewed under the framework of transformations between representations. What we gain from the compositional problem graph perspective is a methodological way to relate together different problems of various forms and complexity, which is especially useful in a lifelong learning setting: the knowledge required to solve one problem is composed of the knowledge required to solve subproblems seen in the past in the context of different problems. For example, we can view latent variable reinforcement learning architectures such as (Ha & Schmidhuber, 2018; Nair et al., 2018) as simultaneously solving an image reconstruction problem and an action prediction problem, both of which share the same subproblem of transforming a visual observation into a latent representation. Lifelong learning, then, can be formulated as not only modifying the connections between nodes in the compositional problem graph but also continuing to make more connections between nodes, gradually expanding the frontier of nodes explored. Sec. 4 describes how CRL takes advantage of this compositional formulation in a multi-task zero-shot generalization setup to solve new problems by re-using computations learned from solving past problems.

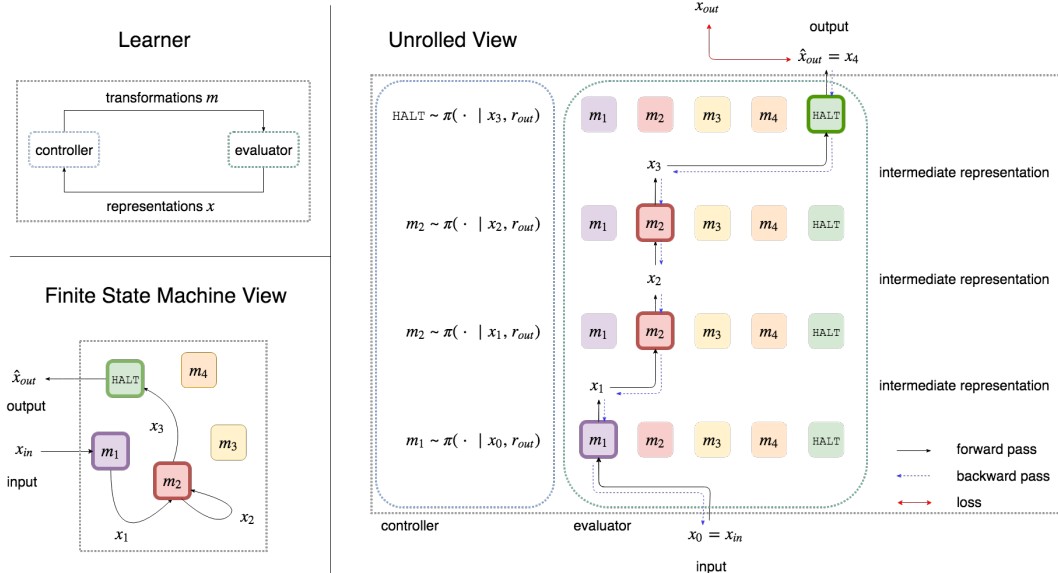

Figure 2: **Compositional recursive learner (CRL):** *top-left*: CRL is a symbiotic relationship between a controller and evaluator: the controller selects a module $m$ given an intermediate representation $x$ and the evaluator applies $m$ on $x$ to create a new representation. *bottom-left*: CRL learns dynamically learns the structure of a program customized for its problem, and this program can be viewed as a finite state machine. *right*: A series of computations in the program is equivalent to a traversal through a Meta-MDP, where module can be reused across different stages of computation, allowing for recursive computation.

**Evaluation.** To evaluate a learner's capacity for compositional generalization, we introduce two challenges. The first is to generalize to problems with different subproblem combinations from what the learner has seen. The second is to generalize to problems with longer subproblems combinations than the learner has seen. Evaluating a learner's capability for compositional generalization is one way to measure how readily old knowledge can be reused and hence built upon.

# 3 A LEARNER THAT PROGRAMS ITSELF

This paper departs from the popular *representation-centric* view of knowledge (Bengio et al., 2013) and instead adopts a *computation-centric* view of knowledge: our goal is to encapsulate useful functionality shared across tasks into specialized *computational modules* – atomic function operators that perform transformations between representations. This section introduces the compositional recursive learner (CRL), a framework for training modules to capture primitive subproblems and for composing together these modules as subproblem solutions to form a path between nodes of the compositional problem graph.

## 3.1 COMPOSITIONAL RECURSIVE LEARNER

The CRL framework consists of a controller $\pi$, a set of modules $m \in M$, and an evaluator $E$. Training CRL on a diverse compositional problem distribution produces a modular recursive program that is trained to transform the input $X_{in}$ into its output $X_{out}$, the corresponding samples of which are drawn from pairs of nodes in the compositional problem graph. In this program, the controller looks at the current state $x_i$ of the program and chooses a module $m$ to apply to the state. The evaluator executes the module on that state to produce the next state $x_{i+1}$ of the program. $X_{in}$ is the initial state of the program, $\hat{X}_{out}$ is the last, and the intermediate states $X_i$ of the execution trace correspond to the other representations produced and consumed by the modules. The controller can choose to re-use modules across different program executions to solve different problems, making it straightforward to re-use computation learned from solving other problems to solve the current one. The controller can also choose to reuse modules several times within the same program execution, which produces recursive behavior.

### 3.2 Deciding Which Computations To Execute

The sequential decision problem that the controller solves can be formalized as a meta-level Markov decision process (meta-MDP) (Hay et al., 2014), whose state space corresponds to the intermediate states of computation $X$, whose action space corresponds to the modules $M$, and whose transition model corresponds to the evaluator $E$. The symbiotic relationship among these components is shown in Fig. 2. In the *bounded-horizon* version of CRL (Sec. 4.2), the meta-MDP has a finite horizon whose length is determined by the complexity of the current problem. In the *infinite-horizon* version of CRL (Sec. 4.1), the program itself determines when to halt when the controller selects the HALT signal. When the program halts, in both versions the current state of computation is produced as output $\hat{x}_{out}$, and CRL receives a terminal reward that reflects how $\hat{x}_{out}$ matches the desired output $x_{out}$. The infinite-horizon CRL also incurs a cost for every computation it executes to encourage it to customize its complexity to the problem.

Note the following key characteristics of CRL. First, unlike standard reinforcement learning setups, the state space and action space can vary in dimensionality across and within episodes because CRL trains on problems of different complexity, reducing more complex problems to simpler ones (Sec. 4.1). Second, because the meta-MDP is internal to CRL, the controller shapes the meta-MDP by choosing which modules get trained and the meta-MDP in turn shapes the controller through its non-stationary state-distribution, action-distribution, and transition function. Thus CRL simultaneously designs and solves reinforcement learning problems "in its own mind," whose dynamics depend just as much on the intrinsic complexity of the problem as well as the current problem-solving capabilities of CRL.

### 3.3 Making Analogies in the Compositional Problem Graph

The solution that we want CRL to discover lies between two extremes, both of which have their own drawbacks. One extreme is where CRL learns a module specialized for every pair of nodes in the compositional problem graph, and the other is where CRL only learns one module for all pairs of nodes. Both extremes yield a horizon-one meta-MDP and are undesirable for compositional generalization: the former does not re-use past knowledge and the latter cannot flexibly continuously learn without suffering from negative transfer.

What is the best solution that CRL could discover? For a given compositional problem graph, an optimal solution would be to recover the original compositional problem graph such that the modules exactly capture the subproblems and the controller composes these modules to reflect how the subproblems were originally generated. By learning both the parameters of the modules and the controller that composes them, during CRL would construct its own internal representation of the problem graph, where the functionality of the modules produces the nodes of the graph. How can we encourage CRL's internal graph to reflect the original compositional problem graph?

We want to encourage the modules to capture the most primitive subproblems, such that they can be composed as atomic computations for other problems. To do this, we need to enforce that each module only has a *local* view of the global problem. If tasks are distinguished from each other based on the input (see Sec. 4.2), we can use domain knowledge to restrict the representation vocabulary and the function class of the modules. If we have access to a task specification (e.g. goal or task id) in addition to the input, we can additionally give only the controller access to the task specification while hiding it from the modules. This forces the modules to be task agnostic, which encourages that they learn useful functionality that generalizes across problems.

Because the the space of subproblem compositions is combinatorially large, we use a curriculum to encourage solutions for the simpler subproblems to converge somewhat before introducing more complex problems, for which CRL can learn to solve by composing together the modules that had been trained on simpler problems. Lastly, to encourage the controller to generalize to new node combinations it has not seen, we train on a diverse distribution of compositional problems, such that the controller does not overfit to any one problem. This encourages controller to make analogies between problems during training by re-using partial solutions learned while solving other problems. Our experiments show that this analogy-making ability helps with compositional generalization because the controller solves new or more complex subproblem combinations by re-using modules that it learned during training.

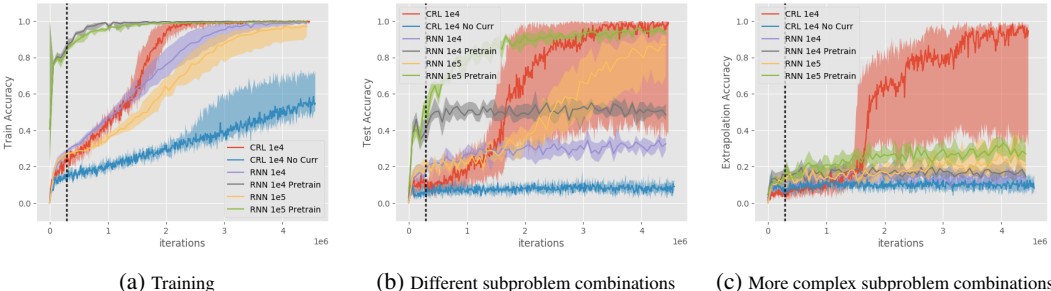

(a) Training      (b) Different subproblem combinations      (c) More complex subproblem combinations

**Figure 3: Multilingual Arithmetic (Quantitative).** CRL generalizes significantly better than the RNN, which, even with ten times more data, does not generalize to 10-length multilingual arithmetic expressions. Pretraining the RNN on domain-specific auxiliary tasks does not help the 10-length case, highlighting a limitation of using monolithic learners for compositional problems. By comparing CRL with a version trained without a curriculum ("No Curr": blue), we see the benefit of slowly growing the complexity of problems throughout training, although this benefit does not transfer to the RNN. The vertical black dashed line indicates at which point all the training data has been added when CRL is trained with a curriculum (red). The initial consistent rise of the red training curve before this point shows CRL exhibits forward transfer (Lopez-Paz et al., 2017) to expressions of longer length. Generalization becomes apparent only after a million iterations after all the training data has been added. **(b, c)** only show accuracy on the expressions with the maximum length of those added so far to the curriculum. "1e4" and "1e5" correspond to the order of magnitude of the number of samples in the dataset, of which 70% are used for training. 10, 50, and 90 percentiles are shown over 6 runs.

# 4 EXPERIMENTS

The main purpose of our experiments is to test the hypothesis that explicitly decomposing a learner around the structure of a compositional problem distribution yields significant generalization benefit over the standard paradigm of training a single monolithic architecture on the same distribution of problems. To evaluate compositional generalization, we select disjoint subsets of node pairs for training and evaluating the learner. Evaluating on problems distinct from those in training tests the learner's ability to *apply* what it has learned to new problems. To demonstrate the broad applicability of the compositional graph, we consider the structured symbolic domain of multilingual arithmetic and the underconstrained and high-dimensional domain of transformed-MNIST classification. We find that composing representation transformations with CRL achieves significantly better generalization when compared to generic monolithic learners, especially when the learner needs to generalize to problems with longer subproblem combinations than those seen during training.

In our experiments, the controller and modules begin as randomly initialized neural networks. The loss is backpropagated through the modules, which are trained with Adam (Kingma & Ba, 2014). The controller receives a sparse reward derived from the loss at the end of the computation, and a small cost for each computational step. The model is trained with proximal policy optimization (Schulman et al., 2017).

## 4.1 MULTILINGUAL ARITHMETIC

This experiment evaluates the infinite-horizon CRL in a multi-objective, variable-length input, symbolic reasoning multi-task setting. A task is to simplify an arithmetic expression expressed in a *source* language, encoded as variable-length sequences of one-hot tokens, and produce the answer modulo 10 in a given *target* language. To evaluate compositional generalization, we test whether, after having trained on 46200 examples of 2, 3, 4, 5-length expressions ($2.76 \cdot 10^{-4}$ of the training distribution) involving 20 of the $5 \times 5 = 25$ pairs of five languages, the learner can generalize to 5-length and 10-length expressions involving the other five held-out language pairs (problem space: $4.92 \cdot 10^{15}$ problems). To handle the multiple target languages, the CRL controller receives a one-hot token for the target language at every computational step additional to the arithmetic expression. The CRL modules consist of two types of feedforward networks: reducers and translators, which do not know the target language and so can only make local progress on the global problem. Reducers transform a consecutive window of three tokens into one token, and translators transform all tokens in a sequence by the same transformation. The CRL controller also selects where in the arithmetic

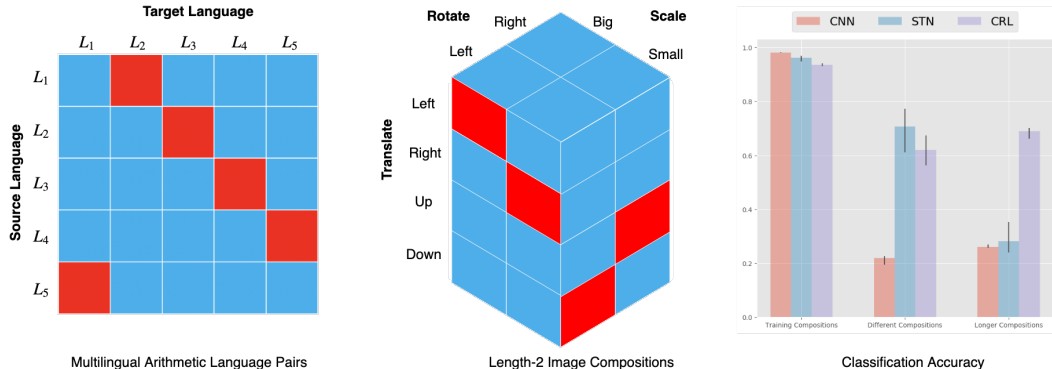

Figure 4: **Left:** For multilingual arithmetic, blue denotes the language pairs for training and red denotes the language pairs held out for evaluation in Fig 3b,c. **Center:** For transformed MNIST classification, blue denotes the length-2 transformation combinations that produced the input for training, red denotes the length-2 transformation combinations held out for evaluation. Not shown are the more complex length-3 transformation combinations (scale then rotate then translate) we also tested on. **Right:** For transformed MNIST classification, each learner performs better than the others in a different metric: the CNN performs best on the training subproblem combinations, the STN on different subproblem combinations of the same length as training, and CRL on longer subproblem combinations than training. While CRL performs comparably with the others in the former two metrics, CRL's $\sim 40\%$ improvement for more complex image transformations is significant.

expression to apply a reducer. We trained by gradually increasing the complexity of arithmetic expressions from length two to length five.

Quantitive results in Fig. 3 show that CRL achieves significantly better compositional generalization than a recurrent neural network (RNN) baseline (Cho et al., 2014) trained to directly map the expression to its answer, even when the RNN has been pretrained or receives 10x more data. Fig. 9 shows that CRL achieves about $60\%$ accuracy for extrapolating to 100-term problems (problem space: $4.29 \cdot 10^{148}$).

The curriculum-based training scheme encourages CRL to designs its own edges and paths to connect nodes in the compositional problem graph, solving harder problems with the solutions from simpler ones. It also encourages its internal representations to mirror the external representations it observes in the problem distribution, even though it has no direct supervision to do so. However, while this is often the case, qualitative results in Fig. 5 show that CRL also comes up with its own *internal* language – hybrid representations that mix different external representations together – to construct compositional solutions for novel problems. Rather than learn translators and reducers that are specific to single input and output language pair as we had expected, the modules, possibly due to their nonlinear nature, tended to learn operations specific to the output language only.

## 4.2 IMAGE TRANSFORMATIONS

This experiment evaluates the bounded-horizon CRL in a single-objective, latent-structured, high-dimensional multi-task setting. A task is to classify an MNIST digit, where the MNIST digit has been randomly translated (left, right, up, down), rotated (left, right), and scaled (small, big). Suppose CRL has knowledge of what untransformed MNIST digits look like; is it possible that CRL can learn to compose appropriate spatial affine transformations in sequence to convert the transformed MNIST digit into a "canonical" one, such that it can use a pre-trained classifier to classify it? To reformulate a scenario to one that is more familar is characteristic of compositional generalization humans: humans view an object at different angles yet understand it is the same object; they may have an accustomed route to work, but can adapt to a detour if the route is blocked. To evaluate compositional generalization, we test whether, having trained on images produced by combinations of two spatial transformations, CRL can can generalize to different length-2 combinations as well as length-3 combinations. A challenge in this domain is that the compositional structure is latent, rather than apparent in the input for the learner to exploit.

CRL is initialized with four types of modules: a Spatial Transformer Network (STN) (Jaderberg et al., 2015) parametrized to only rotate, an STN that only scales, an STN that only translates, and

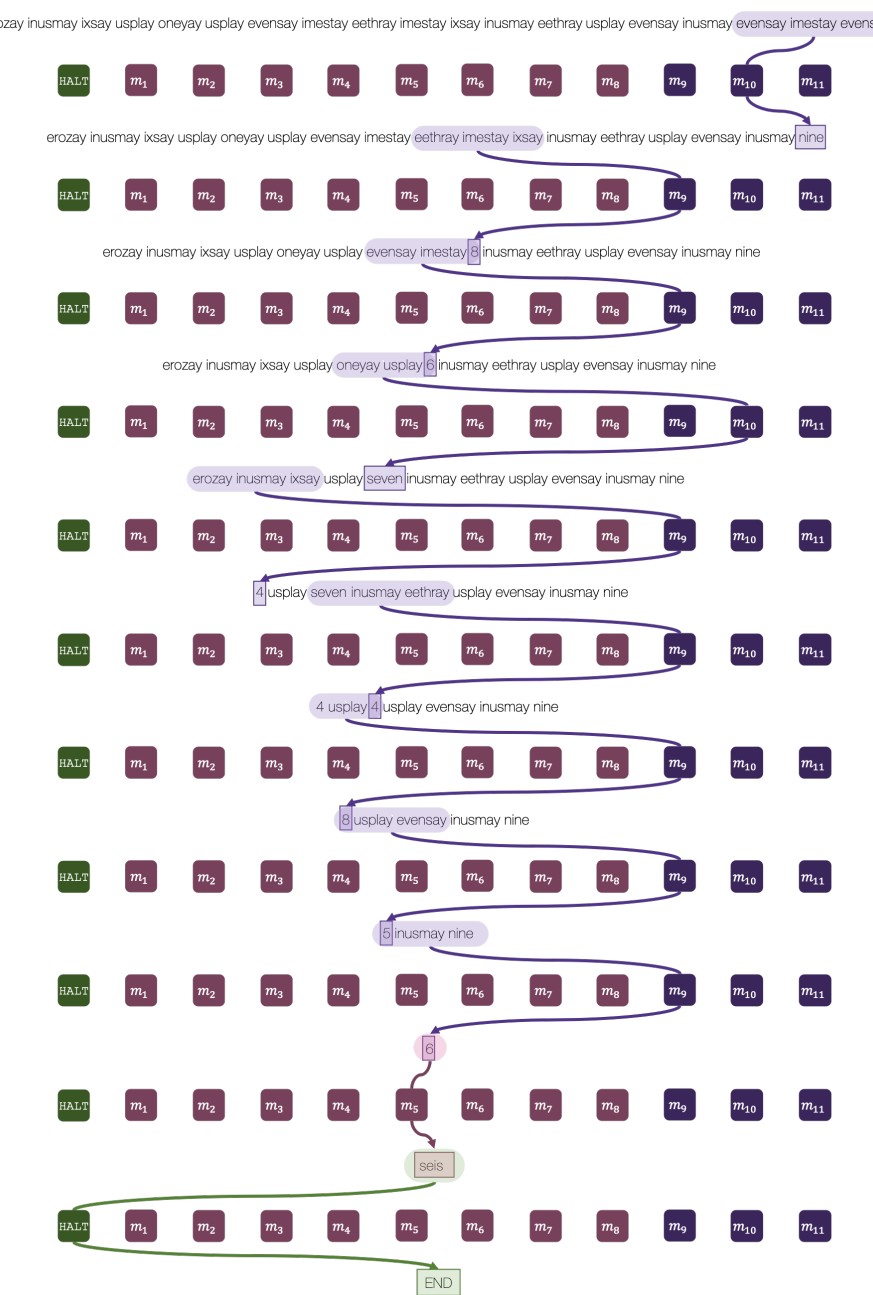

Figure 5: **Multilingual Arithmetic (Qualitative).** A randomly selected execution trace for generalizing from length-5 to length-10 expressions. The input is $0 - 6 + 1 + 7 \times 3 \times 6 - 3 + 7 - 7 \times 7$ expressed in Pig Latin. The desired output is *seis*, which is the value of the expression, 6, expressed in Spanish. The purple modules are reducers and the red modules are translators. The input to a module is highlighted and the output of the module is boxed. The controller learns order of operations. Observe that reducer $m_9$ learns to reduce to numerals and reducer $m_{10}$ to English terms. The task-agnostic nature of the modules forces them to learn transformations that the controller would commonly reuse across problems. Even if the problem may not be compositionally structured, such as translating Pig Latin to Spanish, CRL learns to design a compositional solution (Pig Latin to Numerals to Spanish) from previous experience (Pig Latin to Numerals and Numerals to Spanish) in order to generalize: it first reduces the Pig Latin expression to a numerical evaluation, and then translates that to its Spanish representation using the translator $m_6$. Note that all of this computation is happening internally to the learner, which computes on softmax distributions over the vocabulary; for visualization we show the token of the distribution with maximum probability.

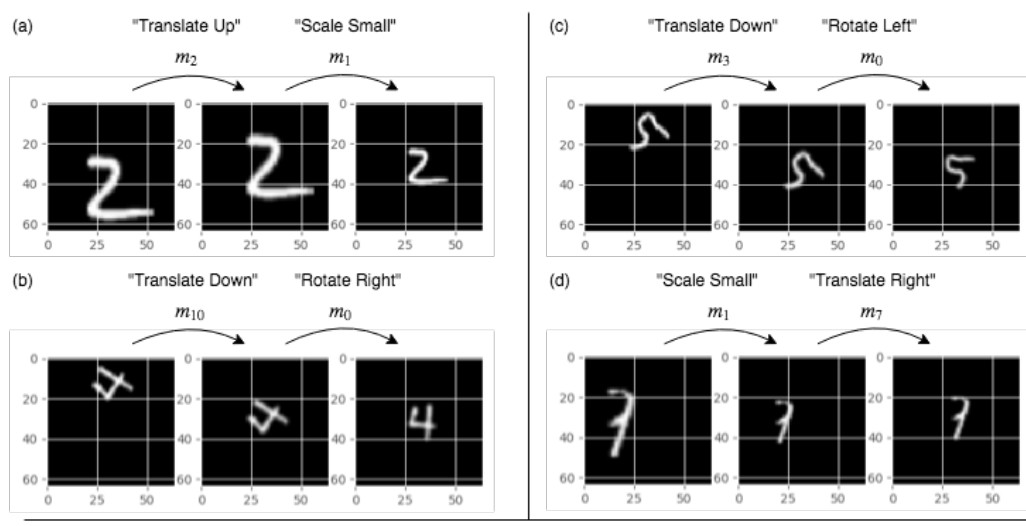

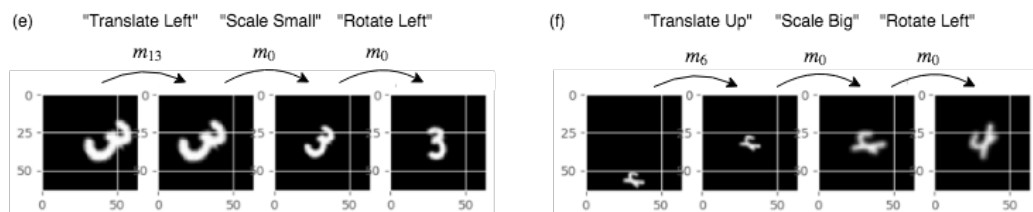

Figure 6: **Image Transformations:** CRL reasonably applies a sequence of modules to transform a transformed MNIST digit into canonical position, and generalizes to different and longer compositions of generative transformations. $m_0$ is constrained to output the sine and cosine of a rotation angle, $m_1$ is constrained to output the scaling factor, and $m_2$ through $m_{13}$ are constrained to output spatial translations. Some modules like $m_2$ and $m_6$ learn to translate up, some like $m_3$ and $m_{10}$ learn to translate down, some like $m_7$ learn to shift right, and some like $m_{13}$ learn to shift left. Consider (d): the original generative transformations were "scale big" then "translate left," so the correct inversion should be "translate right" then "scale small." However, CRL chose to equivalently "scale small" and then "translate right." CRL also creatively uses $m_0$ to scale, as in (e) and (f), even though its original parametrization of outputting sine and cosine is biased towards rotation.

an identity function. All modules are initialized to perform the identity transformation, such that symmetry breaking (and their eventual specialization) is due to the stochasticity of the controller.

Quantitative results in Fig. 4 show that CRL achieves significantly better compositional generalization than both the standard practice of finetuning the convolutional neural network (Springenberg et al., 2014) pretrained classifier and training an affine-STN as a pre-processor to the classifier. Both baselines perform better than CRL on the training set, and the STN's inductive bias surprisingly also allows it to generalize to different length-2 combinations. However, both baselines achieve only less than one-third of CRL's generalization performance for length-3 combinations, which showcases the value of explicitly decomposing problems. Note that in Fig. 6 the sequence of transformations CRL performs are not necessarily the reverse of those that generated the original input, which shows that CRL has learned its own *internal* language for representing nodes in the problem graph.

## 5    RELATED WORK

Several recent and contemporaneous work (Lake & Baroni, 2017; Liška et al., 2018; Loula et al., 2018; Bahdanau et al., 2018) have tested in whether neural networks exhibit *systematic compositionality* (Fodor & Pylyshyn, 1988; Marcus, 1998; Fodor & Lepore, 2002; Marcus, 2018; Calvo & Symons, 2014) in parsing symbolic data. This paper draws inspiration from and builds upon re-

search in several areas to propose an approach towards building a learner that exhibits compositional generalization. We hope this paper provides a point of unification among these areas through which further connections can be strengthened.

## 5.1 COMPOSITIONAL GENERALIZATION

**Transformations between representations:** Our work introduces a learner that exhibits compositional generalization in some sense by bridging deep learning and *reformulation*, or re-representing a problem to make it easier to solve (Holte & Choueiry, 2003; Simon, 1969; Anderson, 1990) by making *analogies* (Oh et al., 2017) to previously encountered problems. Taking inspiration from *meta-reasoning* (Russell & Wefald, 1991; Hay et al., 2014; Hamrick et al., 2017; Graves, 2016) in humans (Griffiths et al., 2015; Callaway et al., 2017; Lieder et al., 2017), CRL generalize to new problems by composing representation transformations (analogous to the subprograms in Schmidhuber (1990)), an approach for which recent and contemporaneous work (Schlag & Schmidhuber, 2018; Alet et al., 2018; Devin et al., 2017) provide evidence.

**Meta-learning:** Our modular perspective departs from recent work in *meta-learning* (Thrun & Pratt, 2012; Schmidhuber, 1987) which assume that the shared representation of monolithic architectures can be shaped by the diversity of tasks in the training distribution as good initializations for future learning (Finn et al., 2017; Nichol et al., 2018; Ravi & Larochelle, 2016; Andrychowicz et al., 2016; Grant et al., 2018; Mishra et al., 2018; Lake et al., 2015; Frans et al., 2017; Gupta et al., 2018b;a; Srinivas et al., 2018).

**Graph-based architectures:** Work in graph-based architectures have studied *combinatorial generalization* in the context of modeling physical systems (Battaglia et al., 2018; Chang et al., 2016; Battaglia et al., 2016; Santoro et al., 2017; Sanchez-Gonzalez et al., 2018; van Steenkiste et al., 2018). Whereas these works focus on factorizing *representations*, we focus on factorizing the *computations* that operate on representations.

## 5.2 NEURAL PROGRAM INDUCTION:

Just as the motivation behind disentangled representations (Whitney et al., 2016; Kulkarni et al., 2015; Chen et al., 2016; Thomas et al., 2017; Bengio et al., 2013; Higgins et al., 2018) is to uncover the latent factors of variation, the motivation behind disentangled programs is to uncover the latent organization of a task. Compositional approaches (as opposed to memory-augmented (Graves et al., 2014; Sukhbaatar et al., 2015; Joulin & Mikolov, 2015; Grefenstette et al., 2015; Kurach et al., 2015; Andrychowicz et al., 2016; Graves et al., 2016) or monolithic (Zaremba & Sutskever, 2014; Kaiser & Sutskever, 2015) approaches for learning programs) to the challenge of discovering reusable primitive transformations and their means of composition generally fall into two categories. The first assumes pre-specified transformations and learns the structure (from dense supervision on execution traces to sparse-rewards) (Reed & De Freitas, 2015; Cai et al., 2017; Xu et al., 2017; Chen et al., 2017; Ganin et al., 2018; Bunel et al., 2018; Feser et al., 2016; Džeroski et al., 2001; Zaremba et al., 2016; Schmidhuber, 1990). The second learns the transformations but pre-specifies the structure (Andreas et al., 2016; Riedel et al., 2016; Lin & Lucey, 2017). These approaches are respectively analogous to our hardcoded-functions and hardcoded-controller ablations in Fig. 7. The closest works to ours from a program induction perspective are (Gaunt et al., 2016; Valkov et al., 2018), both neurosymbolic approaches for learning differentiable programs integrated in a high-level programming language. Our work complements theirs by casting the construction of a program as a reinforcement learning problem, and we believe that more tightly integrating CRL with types and combinators would be an exciting direction for future work.

## 5.3 SELF-ORGANIZING LEARNERS

**Lifelong Learning:** CRL draws inspiration from work (Schmidhuber, 1987; Dechter et al., 2013; Schmidhuber, 2009; 2012; Ellis et al., 2018) on learners that learn to design their own primitives and subprograms for solving an increasingly large number of tasks. The simultaneous optimization over the the continuous function parameters and their discrete compositional structure in CRL is inspired by the interplay between abstract and concrete knowledge that is hypothesized to characterize cognitive development: abstract structural priors serve as a scaffolding within which concrete,

domain-specific learning takes place (Spelke, 1990; Pinker, 1994), but domain-specific learning about the continuous semantics of the world can also provide feedback to update the more discrete structural priors (Gopnik & Wellman, 2012; Carey, 2015).

**Hierarchy:** Several works have investigated the conditions in which hierarchy is useful for humans (Botvinick et al., 2009; Solway et al., 2014; Sanborn et al., 2018); our experiments show that the hierarchical structure of CRL is more useful than the flat structure of monolothic architectures for compositional generalization. Learning both the controller and modules relates CRL to the hierarchical reinforcement learning literature (Barto & Mahadevan, 2003), where recent work (Bacon et al., 2017; Kulkarni et al., 2016; Frans et al., 2017; Vezhnevets et al., 2017; Nachum et al., 2018) attempting to learn both lower-level policies as well as a higher-level policy that invokes them.

**Modularity:** Our idea of selecting different weights at different steps of computation is related to the fast-weights literature (Schmidhuber, 1992; Ba et al., 2016), but those works are motivated by learning context-dependent associative memory (Hopfield, 1982; Willshaw et al., 1969; Kohonen, 1972; Anderson & Hinton, 2014; Ha et al., 2016) rather than composing representation transformations, with the exception of (Schlag & Schmidhuber, 2017). CRL can be viewed as a recurrent mixture of experts (Jacobs et al., 1991), where each expert is a module, similar to other recent and contemporaneous work (Hinton et al., 2018; Rosenbaum et al., 2018; Kirsch et al., 2018; Fernando et al., 2017) that route through a choices of layers of a fixed-depth architecture for multi-task learning. The closest work to ours from an implementation perspective is Rosenbaum et al. (2018). However, these works do not address the problem of generalizing to more complex tasks because they do not allow for variable-length compositions of the modules. Parascandolo et al. (2017) focuses on a complementary direction to ours; whereas they focus on learning causal mechanisms for a single step, we focus on learning how to compose modules. We believe composing together causal mechanisms would be an exciting direction for future work.

## 6 DISCUSSION

This paper sought to tackle the question of how to build machines that leverage prior experience to solve more complex problems than they have seen. This paper makes three steps towards the solution. First, we formalized the compositional problem graph as a language for studying compositionally-structured problems of different complexity that can be applied on various problems in machine learning. Second, we introduced the compositional generalization evaluation scheme for measuring how readily old knowledge can be reused and hence built upon. Third, we presented the compositional recursive learner, a domain-general framework for learning a set of reusable primitive transformations and their means of composition that reflect the structural properties of the problem distribution. In doing so we leveraged tools from reinforcement learning to solve a program induction problem.

There are several directions for improvement. One is to stabilize the simultaneous optimization between discrete composition and continuous parameters; currently this is tricky to tune. Others are to generate computation graphs beyond a linear chain of functions, and to infer the number of functions required for a family of problems. A major challenge would be to discover the subproblem decomposition without a curriculum and without domain-specific constraints on the model class of the modules.

Griffiths et al. (2019) argued that the efficient use cognitive resources in humans may also explain their ability to generalize, and this paper provides evidence that reasoning about what computation to execute by making analogies to previously seen problems achieves significantly higher compositional generalization than non-compositional monolithic learners. Encapsulating computational modules grounded in the subproblem structure also may pave a way for improving interpretability of neural networks by allowing the modules to be unit-tested against the subproblems we desire them to capture. Because problems in supervised, unsupervised, and reinforcement learning can all be expressed under the framework of transformations between representations in the compositional problem graph, we hope that our work motivates further research for tackling the compositional generalization problem in many other domains to accelerate the long-range generalization capabilities that are characteristic of general-purpose learning machines.

ACKNOWLEDGMENTS

The authors would like to thank the anonymous ICLR reviewers and commenters, Alyosha Efros, Dinesh Jayaraman, Pulkit Agrawal, Jason Peng, Erin Grant, Rachit Dubey, Thanard Kurutach, Parsa Mahmoudieh, Aravind Srinivas, Fred Callaway, Justin Fu, Ashvin Nair, Marvin Zhang, Shubham Tulsiani, Peter Battaglia, Jessica Hamrick, Rishabh Singh, Feras Saad, Michael Janner, Samuel Tenka, Kai-I Shan, David Chang, Mei-ling Hsu, Tony Chang and others in the Berkeley Artificial Intelligence Research Lab for helpful feedback, discussions, and support. The authors are grateful for computing support from Amazon, NVIDIA, and Google. This work was supported in part by the Berkeley EECS Department Fellowship for first-year Ph.D. students, travel funding from Bloomsbury AI, contract number FA8650-18-2-7832 from the Defence Advanced Research Projects Agency (DARPA) under the Lifelong Learning Machines program, contract number FA9550-18-1-0077 from the Air Force Office of Scientific Research (AFOSR), and the National Science Foundation (NSF) Graduate Research Fellowship Program. Any opinions, findings, and conclusions or recommendations expressed in this material are those of the authors and do not necessarily reflect the views of DARPA, AFOSR, or the NSF.

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

## A    DATA

**Numerical arithmetic (Sec. D.1):** The dataset contains arithmetic expressions of $k$ terms where the terms are integers $\in [0, 9]$ and the operators are $\in \{+, \times, -\}$. The number of possible problems is $(10^k)(3^{k-1})$. The learner sees $5810/(2.04 \cdot 10^{14}) = 2.85 \cdot 10^{-11}$ of the training distribution. The number of possible problems in the extrapolation set is $(10^{20})(3^{19}) = 1.16 \cdot 10^{29}$. An input expression is a sequence of one-hot vectors of size 13.

| # Terms | Prob. Space | # Train Samples | Frac. of Prob. Space |
|---|---|---|---|
| 2 | $(10^2)(3^1) = 3 \cdot 10^2$ | 210 | $7 \cdot 10^{-1}$ |
| 3 | $(10^3)(3^2) = 9 \cdot 10^3$ | 700 | $7.78 \cdot 10^{-2}$ |
| 4 | $(10^4)(3^3) = 2.7 \cdot 10^5$ | 700 | $2.6 \cdot 10^{-3}$ |
| 5 | $(10^5)(3^4) = 8.1 \cdot 10^6$ | 700 | $8.64 \cdot 10^{-5}$ |
| 6 | $(10^6)(3^5) = 2.43 \cdot 10^8$ | 700 | $2.88 \cdot 10^{-6}$ |
| 7 | $(10^7)(3^6) = 7.29 \cdot 10^9$ | 700 | $9.60 \cdot 10^{-8}$ |
| 8 | $(10^8)(3^7) = 2.19 \cdot 10^{11}$ | 700 | $3.20 \cdot 10^{-9}$ |
| 9 | $(10^9)(3^8) = 6.56 \cdot 10^{12}$ | 700 | $1.07 \cdot 10^{-10}$ |
| 10 | $(10^{10})(3^9) = 1.97 \cdot 10^{14}$ | 700 | $3.56 \cdot 10^{-12}$ |
| Total | $2.04 \cdot 10^{14}$ | 5810 | $2.85 \cdot 10^{-11}$ |

Table 1: Numerical Arithmetic Dataset

**Multilingual arithmetic (Sec. 4.1):** The dataset contains arithmetic expressions of $k$ terms where the terms are integers $\in [0, 9]$ and the operators are $\in \{+, \cdot, -\}$, expressed in five different languages. With 5 choices for the source language and target language, the number of possible problems is $(10^k)(3^{k-1})(5^2)$. In training, each source language is seen with 4 target languages and each target language is seen with 4 source languages: 20 pairs are seen in training and 5 pairs are held out for testing. The learner sees $46200/(1.68 \cdot 10^8) = 2.76 \cdot 10^{-4}$ of the training distribution. The entire space of possible problems in the extrapolation set is $(10^{10})(3^9)(5^2) = 4.92 \cdot 10^{15}$ out of which we draw samples from the 5 held-out language pairs $\big((10^{10})(3^9)(5) = 9.84 \cdot 10^{14}$ possible$\big)$. An input expression is a sequence of one-hot vectors of size $13 \times 5 + 1 = 66$ where the single additional element is a STOP token (for training the RNN).

| # Terms | Prob. Space | Train Prob. Space | # Train Samples | Frac. of Train Dist. | Frac. of Prob. Space |
|---|---|---|---|---|---|
| 2 | $(10^2)(3^1)(25) = 7.5 \cdot 10^3$ | $(10^2)(3^1)(20) = 6 \cdot 10^3$ | $210 \cdot 20 = 4.2 \cdot 10^3$ | $7 \cdot 10^{-1}$ | $5.6 \cdot 10^{-1}$ |
| 3 | $(10^3)(3^2)(25) = 2.25 \cdot 10^5$ | $(10^3)(3^2)(20) = 1.8 \cdot 10^5$ | $700 \cdot 20 = 1.4 \cdot 10^4$ | $7.78 \cdot 10^{-2}$ | $6.22 \cdot 10^{-2}$ |
| 4 | $(10^4)(3^3)(25) = 6.75 \cdot 10^6$ | $(10^4)(3^3)(20) = 5.4 \cdot 10^6$ | $700 \cdot 20 = 1.4 \cdot 10^4$ | $2.6 \cdot 10^{-3}$ | $2.07 \cdot 10^{-3}$ |
| 5 | $(10^5)(3^4)(25) = 2.02 \cdot 10^8$ | $(10^5)(3^4)(20) = 1.62 \cdot 10^8$ | $700 \cdot 20 = 1.4 \cdot 10^4$ | $8.64 \cdot 10^{-5}$ | $6.91 \cdot 10^{-5}$ |
| Total | $2.09 \cdot 10^8$ | $1.68 \cdot 10^8$ | 46200 | $2.76 \cdot 10^{-4}$ | $2.21 \cdot 10^{-4}$ |

Table 2: Multilingual Arithmetic Dataset

**Spatially transformed MNIST (Sec. 4.2):** The generative process for transforming the standard MNIST dataset to the input the learner observes is described as follows. We first center the 28x28 MNIST image in a 42x42 black background. We have three types of transformations to apply to the image: scale, rotate, and translate. We can scale big or small (by a factor of 0.6 each way). We can rotate left or right (by 45 degrees each direction). We can translate left, right, up, and down, but the degree to which we translate depends on the size of the object: we translate the digit to the edge of the image, so smaller digits get translated more than large digits. Large digits are translated by 20% of the image width, unscaled digits are translated by 29% of the image width, and small digits are translated by 38% of the image width. In total there are $2 + 2 + 4 \times 3 = 16$ individual transformation operations used in the generative process. Because some transformation combinations are commutative, we defined an ordering with which we will apply the generative transformations: scale then rotate then translate. For length-2 compositions of generative transformations, there are scale-small-then-translate $(1 \times 4)$, scale-big-then-translate $(1 \times 4)$, rotate-then-translate $(2 \times 4)$, and scale-then-rotate $(2 \times 2)$. We randomly choose 16 of these 20 for training, 2 for validation, 2 for test, as shown in Figure 4 (center). For length-3 compositions of generative transformations, there are scale-small-then-rotate-then-translate $(1 \times 2 \times 4)$ and scale-big-then-rotate-then-translate $(1 \times 2 \times 4)$. All 16 were held out for evaluation.

## B  LEARNER DETAILS

All learners are implemented in PyTorch (Paszke et al., 2017) and the code is available at `https://github.com/mbchang/crl`.

### B.1  ARITHMETIC

**Baseline:** The RNN is implemented as a sequence-to-sequence (Sutskever et al., 2014) gated recurrent unit (GRU) (Cho et al., 2014).

**CRL Controller:** The controller consists of a policy network and a value function, each implemented as GRUs that read in the input expression. The value function outputs a value estimate for the current expression. For the numerical arithmetic task, the policy network first selects a reducer and then conditioned on that choice selects the location in the input expression to apply the reducer. For the multilingual arithmetic task, the policy first samples whether to halt, reduce, or translate, and then conditioned on that choice (if it doesn't halt) it samples the reducer (along with an index to apply it) or the translator.

**CRL Modules:** The reducers are initialized as a two-layer feedforward network with ReLU non-linearities (Nair & Hinton, 2010). The translators are a linear weight matrices.

### B.2  IMAGE TRANSFORMATIONS

**Baselines:** The CNN is a variant of an all-convolutional network (Springenberg et al., 2014). This was also used as the pre-trained image classifier. The affine-STN predicts all 6 learnable affine parameters as in Jaderberg et al. (2015).

**CRL Controller:** The controller consists of a policy network and a value function, each implemented with the same architecture as the CNN baseline.

**CRL Modules:** The rotate-STN's localization network is constrained to output the sine and cosine of a rotation angle, the scale-STN's localization network is constrained to output the scaling factor, and the translate-STN's localization network is constrained to output spatial translations

## C  EXPERIMENT DETAILS

### C.1  MULTILINGUAL ARITHMETIC

**Training procedure:** The training procedure for the controller follows the standard Proximal Policy Optimization training procedure, where the learner samples a set of episodes, pushes them to a replay buffer, and every $k$ episodes updates the controller based on the episodes collected. Independently, every $k'$ episodes we consolidate those $k'$ episodes into a batch and use it to train the modules. We found via a grid search $k = 1024$ and $k' = 256$. Through an informal search whose heuristic was performance on the training set, we settled on updating the curriculum of CRL every $10^5$ episodes and updating the curriculum of the RNN every $5 \cdot 10^4$ episodes.

**Domain-specific details:** In the case that `HALT` is called to early, CRL treats it as a no-op. Similarly, if a reduction operator is called when there is only one token in the expression, the learner also treats it as a no-op. There are other ways around this domain-specific nuance, such as to always halt whenever `HALT` is called but only do backpropagation from the loss if the expression has been fully reduced (otherwise it wouldn't make sense to compute a loss on an expression that has not been fully reduced). The way we interpret these "invalid actions" is analogous to a standard practice in reinforcement learning of keeping an agent in the same state if it walks into a wall of a maze.

**Symmetry breaking:** We believe that the random initialization of the modules and the controller breaks the symmetry between the modules. For episodes 0 through $k$ the controller still has the same random initial weights, and for episodes 0 through $k'$ the modules still have the same random initial weights. Because of the initial randomness, the initial controller will select certain modules more than others for certain inputs; similarly initially certain modules will perform better than others for certain inputs. Therefore, after $k$ episodes, the controller's parameters will update in a direction that will make choosing the modules that luckily performed better for certain inputs more likely;

similarly, after $k'$ episodes, the modules' parameters will update in a direction that will make them better for the inputs they have been given. So gradually, modules that initially were slightly better at certain inputs will become more specialized towards those inputs and they will also get selected more for those inputs.

**Training objective:** The objective of the composition of modules is to minimize the negative log likelihood of the correct answer to the arithmetic problem. The objective of the controller is to maximize reward. It receives a reward of 1 if the token with maximum log likelihood is that of the correct answer, 0 if not, and $-0.01$ for every computation step it takes. The step penalty was found by a scale search over $\{-1, -0.1, -0.01, -0.001\}$ and $-0.01$ was a penalty that we found balanced accuracy and computation time to a reasonable degree during training. There is no explicit feedback on what the transformations should be and on how they are composed.

## C.2    IMAGE TRANSFORMATIONS

**Training procedure:** The training procedure is similar to the mulitlingual arithmetic case. We update the policy every 256 episodes and the modules everye 64 episodes. We observed that directly training for large translations was unstable, so to overcome this we used a curriculum. The curriculum began without any translation, then increased the direction of translation by $1\%$ of the image width every $3 \cdot 10^4$ episodes until the amount of translation matched $20\%$ of the image width for large digits, $29\%$ of the image width for unscaled digits, and $38\%$ of the image width for small digits. Unlike in the multilingual arithmetic case, during later stages of the curriculum we do not continue training on earlier stages of the curriculum.

**Domain-specific details:** In the bounded-horizon setup, we manually halt CRL according to the length of the generative transformation combinations of the task: if the digit was generated by applying two transformations, then we halt CRL's controller after it selects two modules. Therefore, we did not use a step-penalty in this experiment.

**Symmetry breaking:** The transformation parameters were initialized to output an identity transformation, although the the localization network were randomly initialized across modules, which breaks the symmetry among the modules.

**Training objective:** The objective is to classify a transformed MNIST digit correctly based on the negative log likelihood of the correct classification from a pre-trained classifier. The objective of the controller is to maximize reward. It receives a reward of 1 for a correct classification and 0 if not. There is no explicit feedback on what the transformations should be and on how they are composed.

# D    ADDITIONAL EXPERIMENTS

## D.1    NUMERICAL MATH

The input is a numerical arithmetic expression (e.g. $3 + 4 \times 7$) and the desired output (e.g. 1) is the evaluation of the expression modulo 10. In our experiments we train on a curriculum of length-2 expressions to length-10 expressions, adding new expressions to an expanding dataset over the course of training. The first challenge is to learn from this limited data (only 6510 training expressions) to generalize well to unseen length-10 expressions in the test set ($\approx 2^{14}$ possible). The second challenge is to extrapolate from this limited data to length-20 expressions ($\approx 10^{29}$ possible). We compare with an RNN architecture (Chung et al., 2014) directly trained to map input to output.

Though the RNN eventually generalizes to different 10-length expressions and extrapolates to 20-length expressions (yellow in Fig. 7) with 10 times more data as CRL, it completely overfits when given the same amount of data (gray). In contrast, CRL (red) does not overfit, generalizing significantly better to both the 10-length and 20-length test sets. We believe that the modular disentangled structure in CRL biases it to cleave the problem distribution at its joints, yielding this 10-fold reduction in sample complexity relative to the RNN.

We found that the controller naturally learned windows centered around operators (e.g. $2 + 3$ rather than $\times 4-$), suggesting that it has discovered semantic role of these primitive two-term expressions by pattern-matching common structure across arithmetic expressions of different lengths. Note that CRL's extrapolation accuracy here is not perfect compared to (Cai et al., 2017); however CRL

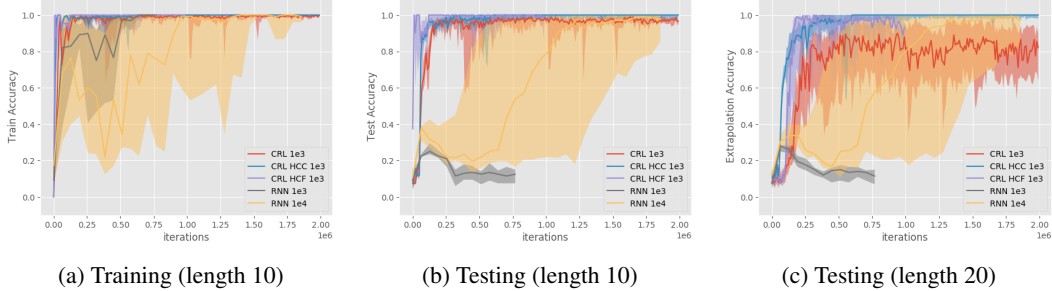

| (a) Training (length 10) | (b) Testing (length 10) | (c) Testing (length 20) |

Figure 7: **Numerical math task.** We compare our learner with the RNN baseline. As a sanity check, we also compare with a version of our learner which has a hardcoded controller (HCC) and a learner which has hardcoded modules (HCF) (in which case the controller is restricted to select windows of 3 with an operator in the middle). All models perform well on the training set. Only our method and its HCC, HCF modifications generalize to the testing and extrapolation set. The RNN requires 10 times more data to generalize to the testing and extrapolation set. For **(b, c)** we only show accuracy on the expressions with the maximum length of those added so far to the curriculum. "1e3" and "1e4" correspond to the order of magnitude of the number of samples in the dataset, of which 70% are used for training. 10, 50, and 90 percentiles are shown over 6 runs.

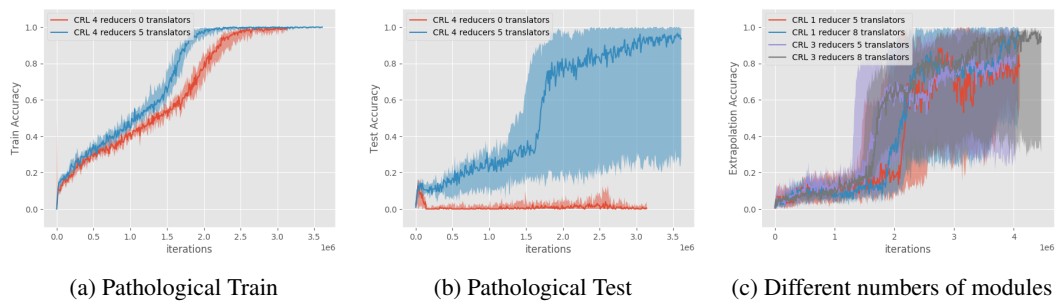

| (a) Pathological Train | (b) Pathological Test | (c) Different numbers of modules |

Figure 8: **Variations:** The minimum number of reducers and translators that can solve the multilingual math problems is 1 and $m$ respectively, where $m$ is the number of languages. This is on an extrapolation task, which has more terms *and* different language pairs. (a, b): Four reducers and zero translators (red) is a pathological choice of modules that causes CRL to overfit, but it does not when translators are provided. (c) In the non-pathological cases, regardless of the number of modules, the learner metareasons about the resources it has to customize its computation to the problem. 10, 50, and 90 percentiles are shown over 6 runs.

achieves such high extrapolation accuracy with only sparse supervision, *without* the step-by-step supervision on execution traces, the stack-based model of execution, and hardcoded transformations.

## D.2 VARIATIONS

Here we study the effect of varying the number of modules available to our learner. Fig. 8a, 8b highlights a particular pathological choice of modules that causes CRL to overfit. If CRL uses four reducers and zero translators (red), it is not surprising that it fails to generalize to the test set: recall that each source language is only seen with four target languages during training with one held out; each reducer can just learn to reduce to one of the four target languages. What is interesting though is that when we add five translators to the four reducers (blue), we see certain runs achieve 100% generalization, even though CRL need not use the translators at all in order to fit the training set. That the blue training curve is slightly faster than the red offers a possible explanation: it may be harder to find a program where each reducer can reduce any source language to their specialized target language, and easier to find programs that involve steps of re-representation (through these translators), where the solution to a new problem is found merely by re-representing that problem into a problem that learner is more familiar with. The four-reducers-five-translators could have overfitted completely like the four-reducers-zero-translators case, but it consistently does not.

We find that when we vary the number of reducers (1 or 3) and the number of translators in (5 or 8) in Fig. 8c, the extrapolation performance is consistent across the choices of different numbers of modules, suggesting that CRL is quite robust to the number of modules in non-pathological cases.

### D.3 How far can we push extrapolation?

Figure 9 shows the extrapolation accuracy from 6 to 100 terms after training on a curriculum from 2 to 5 terms (46200 examples) on the multilingual arithmetic task (Sec. 4.1). The number of possible 100-term problems is $(10^{100})(3^{99})(5^2) = 4.29 \cdot 10^{148}$ and CRL achieves about $60\%$ accuracy on these problems; a random guess would be $10\%$.

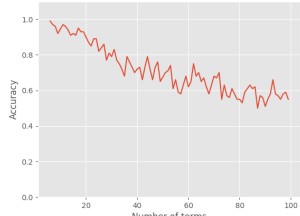

Figure 9: Extrapolation

### D.4 Execution Traces: Function Selection

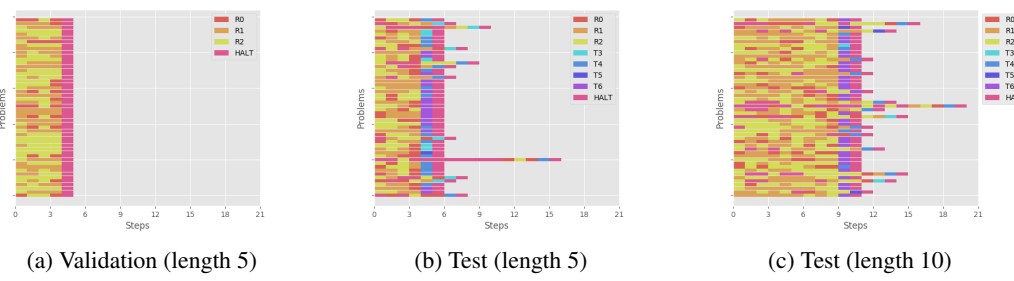

(a) Validation (length 5)  (b) Test (length 5)  (c) Test (length 10)

Figure 10: **Multilingual Arithmetic Execution Traces**

Fig. 10 compares the execution traces of CRL on different language pairs from training of **(a,b)** length 5 and of **(c)** length 10. We observe that in many cases the controller chooses to take an additional step to translate the fully reduced answer into an answer in the target language, which shows that it composes together in a novel way knowledge of how to solve a arithmetic problem with knowledge of how to translate between languages.

### D.5 Execution Traces: Examples

Here are two randomly selected execution traces from the numerical arithmetic extrapolation task (train on 10 terms, extrapolate to 20 terms), where CRL's accuracy hovers around $80\%$. These expressions are derived from the internal representations of CRL, which are softmax distributions over the vocabulary (except for the first expression, which is one-hot because it is the input). The expressions here show the maximum value for each internal representation.

This is a successful execution. The input is `6*1*3-4+6*0*0+1-7-3+3+3*4+1+1+3+3+6+2+7` and the correct answer is 3. Notice that the order in which controller applies its modules does not strictly follow the order of operations but respects the rules of order of operations: for example, it may decide to perform addition (A) before multiplication (B) if it doesn't affect the final answer.

```
6*1*3-4+6*0*0+1-7-3+3+3*4+1+1+3+3+6+2+7     # 3 * 4 = 2
6*1*3-4+6*0*0+1-7-3+3+2+1+1+3+3+6+2+7       # 3 + 2 = 5
6*1*3-4+6*0*0+1-7-3+5+1+1+3+3+6+2+7         # 1 - 7 = 4
6*1*3-4+6*0*0+4-3+5+1+1+3+3+6+2+7           # 0 * 0 = 0
6*1*3-4+6*0+4-3+5+1+1+3+3+6+2+7             # 4 - 3 = 1
6*1*3-4+6*0+1+5+1+1+3+3+6+2+7               # 1 + 3 = 4
6*1*3-4+6*0+1+5+1+4+3+6+2+7                 # 5 + 1 = 6
6*1*3-4+6*0+1+6+4+3+6+2+7                   # 1 + 6 = 7
6*1*3-4+6*0+7+4+3+6+2+7                     # 2 + 7 = 9
6*1*3-4+6*0+7+4+3+6+9                       # 3 + 6 = 9
6*1*3-4+6*0+7+4+9+9                         # 6 * 0 = 0
6*1*3-4+0+7+4+9+9                           # tried to HALT      ----------------------------------------
6*1*3-4+0+7+4+9+9                           # 9 + 9 = 8          everything above this line is extrapolation
6*1*3-4+0+7+4+8                             # 1 * 3 = 3          (A)
6*3-4+0+7+4+8                               # 0 + 7 = 7          (B)
6*3-4+7+4+8                                 # 6 * 3 = 8
8-4+7+4+8                                   # 8 - 4 = 4
4+7+4+8                                     # 4 + 7 = 1
1+4+8                                       # 1 + 4 = 5
5+8                                         # 5 + 8 = 3
3                                           # HALT
END
```

This is an unsuccessful execution trace.

The input is `5+6-4+5*7*3*3*8*0*1-4+6-3*5*3+6-0+0-4-6` and the correct answer is `0`. Notice that it tends to follow of order of operations by doing multiplication first, although it does make mistakes (D), which in this case was the reason for its incorrect answer. Note that CRL never receives explicit feedback about its mistakes on what its modules learn to do or the order in which it applies them; it only receives a sparse reward signal at the very end. Although (C) was a calculation mistake, it turns out that it does not matter because the subexpression would be multiplied by 0 anyways.

```
5+6-4+5*7*3*3*8*0*1-4+6-3*5*3+6-0+0-4-6      # 3 * 8 = 4
5+6-4+5*7*3*4*0*1-4+6-3*5*3+6-0+0-4-6        # 0 - 4 = 6
5+6-4+5*7*3*4*0*1-4+6-3*5*3+6-0+6-6          # 5 * 7 = 5
5+6-4+5*3*4*0*1-4+6-3*5*3+6-0+6-6            # 3 * 4 = 4 (mistake)      (C)
5+6-4+5*4*0*1-4+6-3*5*3+6-0+6-6              # tried to HALT
5+6-4+5*4*0*1-4+6-3*5*3+6-0+6-6              # 5 * 4 = 0
5+6-4+0*0*1-4+6-3*5*3+6-0+6-6                # 6 - 6 = 0
5+6-4+0*0*1-4+6-3*5*3+6-0+0                  # 6 - 3 = 3                (D: order of operations mistake)
5+6-4+0*0*1-4+3*5*3+6-0+0                    # tried to HALT
5+6-4+0*0*1-4+3*5*3+6-0+0                    # tried to HALT
5+6-4+0*0*1-4+3*5*3+6-0+0                    # tried to HALT
5+6-4+0*0*1-4+3*5*3+6+0                      # 3 * 5 = 5
5+6-4+0*0*1-4+5*3+6+0                        # 0 * 1 = 0
5+6-4+0*1-4+5*3+6+0                          # 5 * 3 = 5               ------------------------------------------
5+6-4+0*1-4+5+6+0                            # 0 * 1 = 0               everything above this line is extrapolation
5+6-4+0-4+5+6+0                              # tried to HALT
5+6-4+0-4+5+6+0                              # tried to HALT
5+6-4+0-4+5+6+0                              # tried to HALT
5+6-4+0-4+5+6+0                              # tried to HALT
5+6-4+0-4+5+6+0                              # tried to HALT
5+6-4+0-4+5+6+0                              # tried to HALT
5+6-4+0-4+5+6+0                              # tried to HALT
5+6-4+0-4+5+6+0                              # 6 + 0 = 0
5+6-4+0-4+5+6                                # 5 + 6 = 1
5+6-4+0-4+1                                  # 0 - 4 = 6
5+6-4+6+1                                    # 5 + 6 = 1
1-4+6+1                                      # 1 - 4 = 7
7+6+1                                        # 7 + 6 = 3
3+1                                         # 3 + 1 = 4
4                                           # HALT
END
```