# OpenReview forum: "Automatically Composing Representation Transformations as a Means for Generalization"
_ICLR.cc/2019/Conference_

### Official Review · AnonReviewer2 · 2018-11-02
**Interesting approach to compositionally**

**Rating:** 7
**Confidence:** 3

**Review:**

==== Summary ====

This paper proposes a model for learning problems that exhibit compositional and recursive structure, called Compositional Recursive Learner (CRL). The paper approaches the subject by first defining a problem as a transformation of an input representation x from a source domain t_x to a target domain t_y. If t_x = t_y then it is called a recursive problem, and otherwise a translational problem. A composite problem is the composition of such transformations. The key observation of the paper is that many real-world problems can be solved iteratively by either recursively transforming an instance of a problem to a simpler instance, or by translating it to a similar problem which we already know how to solve (e.g., translating a sentence from English to French through Spanish). The CRL model is essentially composed of two parts, a set of differential functions and a controller (policy) for selecting functions. At each step i, the controller observes the last intermediate computation x_i and the target domain t_y, and then selects a function and the subset of x_i to operate on. For each instance, the resulting compositional function is trained via back-propagation, and the controller is trained via policy gradient. Finally, the paper presents experiments on two synthetic datasets, translating an arithmetic expression written in one language to its outcome written in another language, and classifying MNIST digits that were distorted by an unknown random sequence of affine transformations. CRL is compared to RNN on the arithmetic task and shown to be able to generalize both to longer sequences and to unseen language pairs when trained on few examples, while RNN can achieve similar performance only using many more examples. On MNIST, it is qualitatively shown that CRL can usually (but not always) find the sequence of transformations to restore the digit to its canonical form.

==== Detailed Review ====

I generally like this article, as it contains a neat solution to a common problem that builds on and extends prior work. Specifically, the proposed CRL model is a natural evolution of previous attempts at solving problems via compositionally, e.g. Neural Programmer [1] that learns a policy for composing predefined commands, and Neural Module Networks [2] that learns the parameters of shared differential modules connected via deterministically defined structure (found via simple parse tree). The paper contains a careful review of the related works and highlights the similarities and differences from prior approaches. Though the experiments are mostly synthetic, the underlying method seems to be readily applicable to many real-world problems.

However, the true contributions of the paper are somewhat muddied by presenting CRL as more general than what is actually supported by the experiments. More specifically, the paper presents CRL as a general method for learning compositional problems by decomposing them into simpler sub-problems that are automatically discovered, but in practice, a far more limited version of CRL is used in the experiments, and the suggested translational capabilities of CRL, which are important for abstract sub-problem discovery, are not properly validated:

1. In both experiments, the building-block functions are hand-crafted to fit to the prior knowledge on the compositionally of the problem. For the arithmetic task, the functions are limited to operate each step just on a single window of encompassing 3 symbols (e.g., <number> <op> <number>,  <op> <number> <op>) and return a distribution over the possible symbols, which heavily forces the functions to represent simple evaluators for simple expressions of the form <number> <op> <number>. For the distorted MNIST task, the functions are limited to neural networks which choose the parameter of predetermined transformations (scaling, translation, or rotation) of the input. In both cases, CRL did not *found* sub-problems for reducing the complexity of the original instance but just had to *fine tune* loosely predefined sub-problems. Incorporating expert knowledge into the model like so is actually an elegant and useful trick for solving real problems, and it should be emphasized far clearly in the article. The story of “discovering subproblems” should be left for the discussion / future research section, because though it might be a small step towards that goal, it is not quite there yet.
2. The experiments very neatly show how recursive transformations offer a nice framework for simplifying an instance of a problem. However, the translation capabilities of the model are barely tested by the presented experiments, and it can be argued that all transformations used by the model are recursive in both experiments. First, only the arithmetic task has a translation aspect to it, i.e., the task is to read an expression in one language and then output the answer in a different language. Second, this problem is only weakly related to translation because it is possible to translate the symbols independently, word by word, as opposed to written language that has complex dependencies between words. Third, the authors report that in practice proper translation was only used in the very last operation for translating the computed value of the input expression to the requested language, and not as a method to translate one instance that we cannot solve into another that we can. Finally, all functions operate and return on all symbols and not ones limited to a specific language, and so by the paper’s own definition, these are all recursive problems and not translational ones.

In conclusion, I believe this paper should be accepted even with the above issues, mostly because the core method is novel, clearly explained, and appears to be very useful in practice. Nevertheless, I strongly suggest to the authors to revise their article to focus on the core qualities of their method that can be backed by their current experiments, and correctly frame the discussion on possible future capabilities as such.

[1] Reed et al. Neural Programmer-Interpreters. ICLR 2016.
[2] Andreas et al. Neural Module Networks. CVPR 2016.

---

> ### Author Response · Authors · 2018-11-17
> **Response to Reviewer 2**
>
> We thank Reviewer 2 for their constructive review, which helped us improve the paper in the following aspects. We would be happy to incorporate any other suggestions Reviewer 2 may have for the paper.
>
> 1. We have revised Section 3.1 and the introductory paragraph of Section 3 to be more precise about the domain-specific assumptions CRL makes about the problem distribution. In particular, we included a discussion about restricting the representational vocabulary and the functional form of the modules as a way to incorporate as an inductive bias domain-specific knowledge of the problem distribution.
>
> 2. We agree with Reviewer 2 that the “recursive”/”translational” terminology should be clearer. Therefore, we have revised the “Problems” and “The goal” paragraphs in Section 2 to remove the discussion on translational problems and only focus on recursive problems, where the input and output representations are drawn from the same vocabulary.
>
> 3. Further, we agree with and appreciate Reviewer 2’s analysis that our paper is only a first step towards the full general problem of discovering subproblem decomposition. Accordingly we have revised the end of Section 6 (Discussion) to acknowledge this. We also revised “The challenge” paragraph in Section 2 to be more precise that we are not solving the general subproblem decomposition problem, but rather solving the problem of learning to compose partial solutions to subproblems when the general form of the subproblem decomposition of a task distribution is known.

---

### Official Review · AnonReviewer1 · 2018-11-03
**Trying to learn composition**

**Rating:** 9
**Confidence:** 4

**Review:**

This is a good review paper. I am not sure how much it adds to the open question of how to learn representation with high structure.

I would like to see more detail on what is communicated between the controller and the evaluator. Is it a single function selected or a probability distribution that is sent? How does the controller know how many function the evaluator has created? Or visa versa.

There is a penalty for the complexity of the program, is there a penalty for the number of functions generated?

Having just read Hudson and Manning's paper using a separate controller and action/answer generator they make strong use of attention. It is not clear if you use attention? Maybe in that you can operate on a portion of X. What role does attention play in your work?

---

### Official Review · AnonReviewer3 · 2018-11-05
**Well-written paper; second experiment could be made stronger.**

**Rating:** 7
**Confidence:** 2

**Review:**

Summary: This paper is about trying to learn a function from typed input-output data so that it can generalize to test data with an input-output type that it hasn't seen during training. It should be able to use "analogy" (if we want to translate from French to Spanish but don't know how to do so directly, we should translate from French to English and English to Spanish). It should also be able to generalize better by learning useful "subfunctions" that can be composed together by an RL agent. We set up the solution as having a finite number of subfunctions, including "HALT" which signifies the end of computation. At each timestep an RL agent chooses a subfunction to apply to the current representation until "HALT" is chosen. The main idea is we parameterize these subfunctions and the RL agent as neural networks which are learned based on input -output data. RL agent is also penalized for using many subfunctions. The algorithm is called compositional recursive learner (CRL). Both analogy and meaningful subfunctions should arise purely because of this design.

Multilingual arithmetic experiment. I found this experiment interesting although it would be helpful to specify that it is about mod-10 arithmetic. I was very confused for some time since the arithmetic expressions didn't seem to be evaluated correctly. It also seems that it is actually the curriculum learning that helps the most (vanilla CRL doesn't seem to perform very well) although authors do note that such curriculum learning doesn't help the RNN baseline. It also seems that CRL with curriculum doesn't outperform the RNN baseline that much on test data with the same length as training data. The difference is larger when tested on longer sequences. However here, the CRL learning curve seems to be very noisy, presumably due to the RL element. The qualitative analysis illustrates well how the subfunctions specialize to particular tasks (e.g. translation or evaluating a three symbol expression) and how the RL agent successively picks these subfunctions in order to solve the full task.

Image transformations experiment. This experiment feels a bit more artificial although the data is more complicated than in the previous experiment. Also, in some of the examples in Figure 2, the algorithms seems to perform translation (action 2) twice in a row while it seems like this could be achieved by only one translation. How does this perform experimentally in comparison to an RNN (or other baseline)?

I found this paper to be well-written. Perhaps it could be stronger if the "image transformations" experiment quantitatively compared to a baseline. I'm not an expert in this area and don't know in detail how this relates to existing work (e.g. by Rosenbaum et al; 2018).

Edit: change score to 7 in light of revisions and new experiment.

---

> ### Author Response · Authors · 2018-11-17
> **Response to Reviewer 3**
>
> We thank Reviewer 3 for their constructive review, which helped us improve the paper in various aspects. We would be happy to incorporate any other suggestions Reviewer 3 may have for the paper. We would like to make the following clarifications:
>
> 1. We have clarified in Section 4.1 that arithmetic problems are modulo-10.
>
> 2. With regards to how CRL compares to the RNN on test data with the same length as the training data, Figure 2b shows that there is a substantial difference between CRL (red curve) and RNN (purple curve). It is only with 10x more data does the RNN (yellow curve) reach comparable performance with CRL.
>
> 3. Reviewer 3 noted that in the right half of Figure 4, the top-two examples showed that CRL performs transformation twice, when in fact this can be achieved by only translation. This is true. For simplicity, we had fixed the number of transformations to two transformations. That CRL finds alternate ways of achieving the same end representation (using two translations instead of one) illustrates a core feature of the CRL framework: that it is possible to solve a problem (e.g. a large translation) by composing together partial solutions (two small translation).
>
> 4. We will have the baseline experiments Reviewer 3 requested in time for the final, and will endeavor to add these in to the paper during the discussion period.

---

> > ### Author Response · Authors · 2018-12-05
> > **Quantitive Evaluation of MNIST transformations**
> >
> > Below is a quantitative evaluation of how CRL compares with a CNN baseline.
> >
> > The dataset contains MNIST digits that have been scaled (S), rotated (R), and translated (T). There are two types of scaling: large and small. There are two types of rotation: left and right. There are four types of translation: left, right, up, and down. The set of depth-2 compositions (20 total) we considered are scale->translate (2*4 possible), rotate->translate (2*4 possible), scale->rotate (2*2 possible). “scale->translate” means that the image was first scaled, then translated. The set of depth-3 compositions we considered are scale->rotate->translate (2*2*4 possible).
> >
> > The training set is 16 out of the 20 depth-2 compositions, the first hold-out set is the remaining 4 out of the 20 depth-2 compositions, and the second hold-out set is the set of depth-3 compositions. The first hold-out set tests extrapolation to a disjoint set of transformation combinations of the same depth as training; the second hold-out set tests extrapolation to a set of transformation combinations of longer depth than in training.
> >
> > The CNN baseline was pre-trained to classify canonical MNIST digits, and it continued training on transformed MNIST digits.
> > CRL used the same pre-trained MNIST classifier as a decoder (whose weights are frozen), and learned a set of Spatial Transformer Networks (STN) constrained to rotate, scale, or translate.
> > We noticed instability in training the STNs to model drastic translations (where the digit was translated more than 15% the width of the images). A potential reason for this is that because the weights of CRL’s decoder (pre-trained MNIST classifier) are frozen, the classifier acts as a more complex loss functions for the upstream STNs. We addressed this challenge by defining a curriculum for the translated data, where initially the digit was translated by a small amount, and at the end of the curriculum, the digit is translated to the far edge of the image. We applied this curriculum to both CRL and the baseline.
> >
> > The results are as follows (over 5 random seeds):
> >
> > Training set accuracy:
> > ———————————
> > CNN
> > median: 0.98
> > 10% quantile: 0.98
> > 90% quantile: 0.98
> > CRL
> > median: 0.89
> > 10% quantile: 0.87
> > 90% quantile: 0.90
> >
> > Hold-out set (same depth)
> > ———————————
> > CNN
> > median: 0.22
> > 10% quantile: 0.19
> > 90% quantile: 0.23
> > CRL
> > median: 0.67
> > 10% quantile: 0.59
> > 90% quantile: 0.71
> >
> > Hold-out set (longer depth)
> > ———————————
> > CNN
> > median: 0.26
> > 10% quantile: 0.26
> > 90% quantile: 0.27
> > CRL
> > median: 0.69
> > 10% quantile: 0.60
> > 90% quantile: 0.71
> >
> > We notice that CRL performs a bit worse on the training set because it is constrained to go through the bottleneck of only using Spatial Transformation Networks, whereas the CNN is free to fit the training set without such constraints. In the hold-out sets, it is clear that the CNN overfits to the training set and is unable to classify MNIST digits that have been transformed by a set of transformation combinations it has not seen before. CRL, on the other hand, generalizes significantly better because it re-uses the primitive spatial transformations it had learned during training to re-represent the image into a canonical MNIST digit.

---

### Public Comment · (anonymous) · 2018-10-25
**Relation of the Compositional Recursive Learner to Routing Networks**

I have read the paper "Automatically Composing Representation Transformations as a Means for Generalization" with great pleasure. I particularly enjoyed how the paper tries to link compositionality to analogical reasoning. I think an architecture for compositional reasoning that can solve even complex tasks elegantly is of great value.
I do though have some concerns about the relationship between the "Compositional Recursive Learner" (CRL) and "Routing Networks" (RN).  Specifically, it seems to me that the CRL is an example of a single agent recursive routing network, as described in (Rosenbaum et al, ICLR 2018). In particular, the design of a compositional computation and learning framework that combines trainable function blocks with a reinforcement learning meta learner (as described in section 3.2 and 3.3) is highly similar (section 3.2) or nearly identical (section 3.3) to the formulation in the routing networks paper.
The main difference is that while (Rosenbaum et al) focused on a limited-horizon recurrence (see pages 1, 3, 4, 7, and particularly 14 in the appendix), CRL uses an infinite-horizon recurrence.
Surprisingly, this relationship is not discussed in the paper in any detail. Routing Networks are more closely examined in the appendix only. Additionally, there are two stated assumptions (on p. 15) on routing networks that I do not think are true: (1) Routing Networks necessarily have a separate controller per computation step and (2) Routing Networks necessarily use a different set of functions per computation step.  The idea of an RN with a single controller applied across computation steps is discussed on page 5 of (Rosenbaum et al).  The idea of re-using function blocks across computation steps is discussed on pages 1, 3, 4, 7 and 14.

Given the obviously close relationship between these two works, I feel that the connection should be more emphasized and the comparison more central to the paper. And indeed, the results shown for routing networks are somewhat hard to believe (at least for smaller problems as routing networks are not expected to scale to inputs of the same size). Is the routing networks implementation compared to actually also recurrent? Does the routing network receive the same curriculum learning strategy training?

The link to Rosenbaum et al in ICLR 2018: https://openreview.net/forum?id=ry8dvM-R-

---

> ### Public Comment · (anonymous) · 2018-11-03
> **Update to concerns above**
>
> Now that the review period is officially over, I was hoping to get a response to the issues raised above. I ask the authors to address the following questions in particular:
> 1. Do the authors agree with the assessment that the CRL is in effect a Routing Network? (I might point out that the authors even hint at that in the arxiv version of this paper)
> 2. Do the authors agree that the only two minor differences (apart from the training schedule) are (1) that the CRL has infinite horizon recurrence, while RNs only have limited horizon recurrence, and (2) the RL algorithm chosen? (this implies a mischaracterization of RNs on the authors part)
> 3. In light of the previous two points, why do the authors claim that their architecture is novel? (this critique does not extend to the other parts of their paper)

---

> > ### Author Response · Authors · 2018-11-04
> > **Response to "Update to concerns above"**
> >
> > Question 1
> >
> > Although we have acknowledged the similarities in "Response to Relation of the Compositional Recursive Learner to Routing Networks", we respectfully disagree with OP that “CRL is in effect a Routing Network.” To make such a statement would be to mischaracterize the difference between the generative nature of CRL and the routing-based nature of RN and to ignore the respective problem domains that CRL and RN tackles.
> >
> > We focus on the extrapolation problem (Sec 2 and 3), for which learning on multiple tasks is a means to this end, whereas Rosenbaum et al. focus on task interference, for which multi-task learning is the end itself (see abstract of Rosenbaum et al.). Because our focus is on subproblem decomposition, CRL restricts the representation space such that harder problems can be expressed in the same vocabulary as easier problems. RN do not focus on subproblem decomposition, so it is not clear whether their modules learn any interpretable atomic functionality or whether their representations capture semantic boundaries between subproblems that comprise a larger problem. Therefore, RN does not have the inductive bias for extrapolation problems that require the learner to re-represent the new problem in terms of problems the learner has seen during training.
> >
> > The key methodological difference between CRL and RN lies in the generative nature of CRL and the routing-based nature of RN. RN and other work such as PathNet (Fernando et al. 2017) route input-dependent paths through a large fixed architecture. In contrast, the extrapolation problem necessitates CRL be generative, meaning that it incrementally builds module on top of module without a fixed computational horizon. This is necessary for the problem domain we consider, in which we want to train and extrapolate to different problems that require various computation depths. Therefore, variable-length computation horizon, the restrictions on the representational vocabulary, and the emergent semantic functionality of its submodules as solutions to subproblems within a larger problem (see Figure 3) are crucial design considerations for the capability of CRL that RN does not incorporate in their approach.
> >
> > Question 2
> >
> > Based on the crucial difference between the generative nature of CRL and the routing-based nature of RN, the variable computation horizon is a crucial feature of CRL, not a minor difference, as we discussed above and in the Related Work. Because of the variable computation horizon, it is not possible to have a separate controller at each timestep/depth because the number of time steps of computation unknown; therefore this is also not a minor difference.
> >
> > We agree with OP that the particular RL algorithm (PPO vs MARL-WPL) is not particularly relevant to the central focus of our paper, which is extrapolation in compositionally structured problems, and we indeed did not claim so. Nevertheless, our work represents an algorithmic improvement that does make the single controller architecture more effective (above 90% extrapolation accuracy for multilingual arithmetic) than Rosenbaum et al.’s architecture (Figure 4 and Figure 5 of Rosenbaum et al. shows < 50% accuracy, whereas their best method achieves around 60%).
> >
> > CRL’s focus on capturing interpretable atomic functionality in its modules and using representations capture semantic boundaries between subproblems that comprise a larger problem are important ingredients for CRL’s analogical reasoning: literally re-representing a problem in terms of problems it has already seen. This is another key difference between RN and CRL, because the architectural design of RN do not have the inductive bias (restrictions on the modules and representations) that encourage it re-represent problems in literally terms of previously-seen problems.
> >
> > Question 3: Novelty
> >
> > The novelty of our work (with respect to RN) lies in the generative nature of CRL because we reframe of the extrapolation problem as a problem of learning algorithmic procedures over transformations between representations, as discussed in the abstract, intro, and discussion. CRL generates function composition, in contrast to how RN routes through function paths. As shown in the experiments section, the transformations CRL learns have interpretable, atomic functionality and the representations capture semantic boundaries between subproblems that comprise a larger problem. These features of the CRL architecture crucially differentiate it from other routing-based architectures, including RN and PathNet.

---

> ### Author Response · Authors · 2018-11-04
> **Response to "Relation of the Compositional Recursive Learner to Routing Networks"**
>
> We are grateful to the Anonymous Commenter (OP) for their detailed and insightful comment.
>
> It is true, as OP points out, that there is a close connection to Routing Networks (RN), an important and interesting paper that seeks to mitigate task interference in multi-task learning by routing through the modules of a convolutional neural network. Like RN, a feature of our work is that the learner creates and executes a different computation graph for different inputs, where this computation graph consists of a series of functions applied according to a controller. Therefore, it is possible to see CRL as taking a step beyond the single-controller (which they refer to as “single-agent” in Rosenbaum et al.) version of RN by incorporating several algorithmic improvements that make the single controller version not only effective for solving the task (c.f. Figure 4 and Figure 5 of Rosenbaum et al.) but also effective for extrapolation, a problem domain that Rosenbaum et al. does not consider.
>
> We will follow OP’s recommendation and make the comparison with RN more salient in the experiments and related work section. However, we would like to emphasize that the problem that RN tackles (mitigating task interference in multi-task learning) is not the central focus of the paper. That CRL and RN started from significantly different motivations and problem domains but converged to a similar architecture design serves as encouraging evidence in support of an old idea that exploiting modularity and encapsulation yield help more efficiently capture the modalities of a task distribution, and we are excited that both we and Rosenbaum et al. are actively pushing this front.
>
> We thank OP for pointing out it is indeed true that 1) RN does not necessarily have a separate controller per time step and 2) RN does not necessarily use a different set of functions per computation step; we will follow OP’s recommendation and clarify this in the next version of the paper to avoid potential misunderstanding. One source for our misunderstanding is that the exposition of RN in section 3 of Rosenbaum et al. (e.g. “If the number of function blocks differs from layer to layer in the original network, then the router may accommodate this by, for example, maintaining a separate decision function for each depth” (page 4, Rosenbaum et al.) and “The approximator representation can consist of either one MLP that is passed the depth (represented in 1-hot), or a vector of d MLPs, one for each decision/depth” (page 5, Rosenbaum et al.)) seems to heavily suggest the two assumptions we made on page 15 of our manuscript, so we thought that the single-controller or shared function cases were included in Rosenbaum et al. mostly for the sake of comparison. The reason that our submission discussed points (1) and (2) was not intended to misrepresent RN. Rather it was because we interpreted Figure 4, Figure 5, Table 3, Table 4 of Rosenbaum et al. as claiming the routing-all-fc (one-agent-per-task, separate controller per depth, different functions-per-layer) as the flag bearer of their results. To make the comparison that most fairly represents RN’s claims, we had conducted our comparison based on the best version of RN reported in Rosenbaum et al. (routing-all-fc), which uses a separate controller per depth and a different set of functions per depth (according to Table 3 and 4 in Rosenbaum et al.).

---

> > ### Author Response · Authors · 2018-11-17
> > **Comparison with Routing Networks**
> >
> > Based on OP’s suggestions, we have included a paragraph in Section 3.4 (“Discussion of Design Choices”) that features a discussion that compares CRL with Routing Networks.
> >
> > To avoid misrepresenting Routing Networks, we have revised the wording of the experiment of Appendix D.2 to compare with a mixture-of-expert- inspired baseline, rather than Routing Networks, because as OP points out, 1) RN does not necessarily have a separate controller per time step and 2) RN does not necessarily use a different set of functions per computation step. The purpose of this experiment is to show the benefits of reusing modules across computation steps and to show the benefit of allowing a flexible computation horizon.

---

### Meta-Review · Area_Chair1 · 2018-12-11
**Nice framing of the problem; architecturally somewhat incremental over routing nets**

**Confidence:** 5
**Recommendation:** Accept (Poster)

**Metareview:**


pros:
- the paper is well-written and presents a nice framing of the composition problem
- good comparison to prior work
- very important research direction

cons:
- from an architectural standpoint the paper is somewhat incremental over Routing Networks [Rosenbaum et al]
- as Reviewers 2 and 3 point out, the experiments are a bit weak, relying on heuristics such as a window over 3 symbols in the multi-lingual arithmetic case, and a pre-determined set of operations (scaling, translation, rotation, identity) in the MNIST case.

As the authors state, there are three core ideas in this paper (my paraphrase):

(1) training on a set of compositional problems (with the right architecture/training procedure) can encourage the model to learn modules which can be composed to solve new problems, enabling better generalization.
(2) treating the problem of selecting functions for composition as a sequential decision-making problem in an MDP
(3) jointly learning the parameters of the functions and the (meta-level) composition policy.

As discussed during the review period, these three ideas are already present in the Routing Networks (RN) architecture of Rosenbaum et al.  However CRL offers insights and improvements over RN algorithmically in a several ways:

(1) CRL uses a curriculum learning strategy.  This seems to be key in achieving good results and makes a lot of sense for naturally compositional problems.
(2) The focus in RN was on using the architecture to solve multi-task problems in object recognition. The solutions learned in image domains while "compositional" are less clearly interpretable.  In this paper (CRL) the focus is more squarely on interpretable compositional tasks like arithmetic and explores extrapolation.
(3) The RN architecture does support recursion (and there are some experiments in this mode) but it was not the main focus.  In this paper (CRL) recursion is given a clear, prominent role.

I appreciate that the authors' engagement in the discussion period. My feeling is that  the paper offers nice improvements, a useful framing of the problem, a clear recursive formulation, and a more central focus on naturally compositional problems.  I am recommending the paper for acceptance but suggest that the authors remove or revise their contributions (3) and (4) on pg. 2 in light of the discussion on routing nets.

Routing Networks, Adaptive Selection of Non-Linear Functions for Multi-task Learning, ICLR 2018